# Delayed Propagation Transformer:
# A Universal Computation Engine towards Practical Control in Cyber-Physical Systems

**Wenqing Zheng**
The University of Texas at Austin
`w.zheng@utexas.edu`

**Qiangqiang Guo**
University of Washington
`guoqq17@uw.edu`

**Hao Yang**
University of Washington
`haoya@uw.edu`

**Peihao Wang**
The University of Texas at Austin
`peihaowang@utexas.edu`

**Zhangyang Wang**
The University of Texas at Austin
`atlaswang@utexas.edu`

## Abstract

Multi-agent control is a central theme in the *Cyber-Physical Systems (CPS)*. However, current control methods either receive non-Markovian states due to insufficient sensing and decentralized design, or suffer from poor convergence. This paper presents the *Delayed Propagation Transformer* (**DePT**), a new transformer-based model that specializes in the global modeling of CPS while taking into account the immutable constraints from the physical world. DePT induces a cone-shaped spatial-temporal attention prior, which injects the information propagation and aggregation principles and enables a global view. With physical constraint inductive bias baked into its design, our DePT is ready to plug and play for a broad class of multi-agent systems. The experimental results on one of the most challenging CPS – network-scale traffic signal control system in the open world – show that our model outperformed the state-of-the-art expert methods on synthetic and real-world datasets. Our codes are released at: `https://github.com/VITA-Group/DePT`.

## 1 Introduction

The *Cyber-Physical System (CPS)* is ubiquitous in our modern society; examples include intelligent transportation systems, power grids, autonomous automobile systems, industrial control systems, and robotic swarms. A CPS consists of multiple physical agents that interact and cooperate from time to time; as well as a cyberspace that is responsible for monitoring the status of the physical system, predicting future states, and/or assigning control actions to the physical agents [1, 2]. How to effectively execute multi-agent control over CPS is an open problem of vital importance.

In general, there are two types of methods to solve the CPS multi-agent control problem: decentralized and centralized ones. The decentralized methods usually design an individual controller for each agent based on local states. Due to the agents' continuous interaction, the features and the influence of actions propagate between agents, hence the state transition for each agent relies on not only the locally observed states and local actions but also states and actions of other agents. Therefore, the locally observed states alone cannot make up a Markov Decision Process (MDP) [3, 4], which requires the controller to balance the global as well as the local features [5, 6, 7]. Existing methods tackle this issue by either complementing local information with information from the node's immediate neighbors or self history, or using Graph Convolutional Networks (GCNs) instead to recursively model information propagation and aggregation. However, no aforementioned method is free of the

35th Conference on Neural Information Processing Systems (NeurIPS 2021).

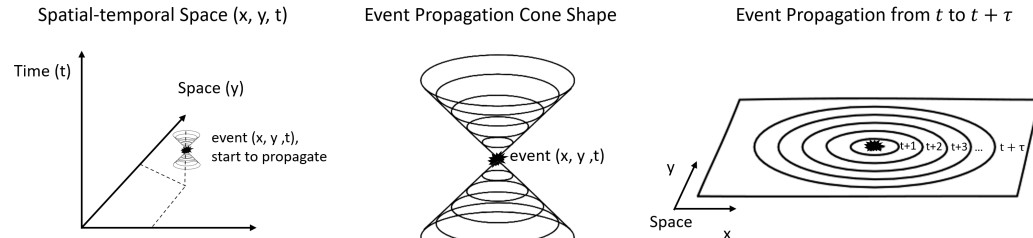

Figure 1: Illustration of the typical impact propagation pattern of an event in a *Cyber-Physical System (CPS)*. Analogous to the "light cone" concept in the special theory of relativity, the correlation pattern in the space-time spans a cone-shaped distribution, which could be set as our prior for DePT.

limited localized horizon issue: an $n$-layer GCNs can view at most $n$-hop neighbors away; meanwhile, deeper GCNs usually suffer from the "bottleneck phenomenon" [8] and "over-smoothness" issues [9], which make GNNs practically hard to train and perform well. Hence, such methods are inherently restricted to making a complete MDP state. In addition, the edges in a CPS graph often have physical directions to propagating information. That caused "feature mismatching" [10, 11] when modelling by normal GCNs, and several methods [12, 13, 14] have studied directed graph modeling as remedies.

In contrast to decentralized methods, centralized methods partially avoided the non-Markovian barrier introduced by mutual interactions. However, due to the incomplete coverage/deployment of sensors and the sensing limitations, the ideal MDP assumption cannot always be met by real-world CPS observations [15, 16]. Moreover, it is difficult for the traditional centralized methods to converge due to the huge state and action spaces [17]. If one considers $N$ agents with $d_S$ dimensional state space and $d_A$ actions per agent, then the state space size is $d_S * N$, and the total number of actions (i.e., number of logits in the output layer of traditional centralized model) will become $(d_A)^N$.

Fortunately, the booming of transformer models shed a new light on addressing the aforementioned difficulties [18, 19, 20, 21, 22, 23, 24, 25, 26]. The Transformer architectures belong to the *centralized methods* due to their global view in the attention mechanism. Their global views free them from the locality inductive bias of GCNs, and make them promising candidates for globally controlling multi-agent CPS problems. Also thanks to the tokenization and *decentralized processing* of each tokens, transformers are also free from the huge state/action space issues: given $d_S$ dimensional state space and $d_A$ actions for each agent, transformer's input/output are still $d_S/d_A$, regardless of the agent quantity $N$. However, the traditional transformer is **not immediately ready to be plugged in** tackling CPS control problems due to the following two arising challenges:

**#1. Lack of physics in the attention modeling.** The classical transformers benefit from their fully flexible self-attention mechanisms to capture the complex interactions within data. This free-form is desirable for vision and NLP tasks [27, 28], but no longer valuable when it comes to real CPS due to its unawareness of many physical constraints. One example is the directional propagation and direction-feature coupling issue previously mentioned. For another prominent example, the real passage of information flow between nodes and the propagation of the effect of past actions are subject to the physical propagation speed. Previous studies about graph learning using transformers mainly focus on non-physical systems and encourage between-node communication without any notion of physical latency [29, 30]. In CPS, however, features take time to propagate through the physical world. Without that important physical delay constraint in mind, the learned self-attention might mislead the attention to non-relevant temporal events or physically disjoint node pairs.

**#2. The difficulty of training and convergence under noisy CPS data**. Transformers are strong universal representers free of inductive bias, such as those in convolutions or recurrence. But the blessing of unprecedented flexibility and larger capacity can turn into the curse at training: transformers take a much longer time to converge. They are resource-heavier and much more data-hungry compared to classical convolutional, or recurrent networks [24, 26]. In CPS control problems, the training data will suffer from even higher variance due to the unavoidable randomness in state transitions. Such issue is further amplified in the multi-agent setting. How to properly train transformers to stable convergence in this scenario can still present a daunting challenge.

In view of the above, we propose to model the delayed propagation effect in CPS by enforcing the physical constraints into the transformer, via a *cone-shaped temporal-spatial prior* – that the features propagation in physical world process naturally forms the cone shape in the time-space [31, 32]. We customized a multi-level heterogeneous attention mechanism that has the cone shaped prior baked into its design. With such design, the attention between input tokens now have a physical interpretation, hence guiding the controller to learn more effectively in exploring the collaborative strategy across agents. Our main **technical innovations** are summarized below:

- We model the CPS control problem under a transformer-based framework. This is a naturally motivated step: transformers are free from the locality restrictions, and learn inter-agent correlations across the whole system with constant state/action space size.
- We propose the ***Delayed Propagation Transformer (DePT)***, a new type of transformer with spatio-temporal priors baked into the attention design, to better characterize the inductive bias of physical information propagation latency, accompanied with better interpretability.
- We build a CPS controller based on DePT, and take the well-received CPS benchmark – transportation signal control system as a study case. The proposed controller achieves the state-of-the-art performance in the challenging urban scale traffic signal control task.

## 2   Related Works

**CPS Control and Learning.**   With the rise of Reinforcement Learning (RL), learning-based control systems are adapted into CPS systems, including traffic network [33, 34], smart gird system [35, 36], and autonomous vehicle [37]. Using traffic network control as an example, IntelliLight [38] developed a deep RL model with policy interpretations. PressLight [39] considered a multi-intersection problem, where an RL agent was trained for every individual intersection. [40] introduced a fully localized RL agent to control traffic signals at every intersection. To efficiently model the influence of the neighbor intersections, a graph attentional network was further introduced in CoLight [41]. [42] reviewed conventional and RL-based methods for TSCP in detail.

To boost the optimization process and enhance the model transferability, [43] proposed to speed up the learning process by leveraging the knowledge learned from existing scenarios. To investigate the cooperation mechanism among various types of nodes and enhance the model utility, AttendLight [44] designed a multi-level attention mechanism to handle various numbers of roads-lanes, thus enabling decision-making with different phases in an intersection. However, the previous control scheme in CPS, especially for traffic network control, either assumed idealistic sensing condition, or tended to downplay or even ignore inter-agent cooperation and global signal propagation.

**Transformers.**   Transformers have witnessed great success in the application of natural language processing [24, 18], and recently in computer vision too [21, 45, 46]. In general, researchers use transformers to process the node embedding in two orthogonal directions: first, through the node-wise residual feature transformation, an arbitrary type of intra-node transformation is enabled [18, 47, 48]; second, through the attention mechanism, features from different nodes are dynamically aggregated and the inter-nodes relationships are captured [48]. Previous efforts have shown the potential of transformers in multi-agent system [49], by flattening connections features across time and agents. [50] used transformers to tackle the sparse communication in multi-agent settings, and achieved state-of-the-art (SOTA) performance. [30] introduced transformer for the graph-to-sequence learning in translation, which also inspired us to use explicit relation encoding to allow direct cooperation between two distant nodes in agent features sharing. Yet to our best knowledge, existing transformers have not explicitly tackled the spatial-temporal features together with the connections encoding, and no effort has been devoted to their learning under CPS physical constraints.

## 3   Problem Settings

A general CPS can be modeled by a graph $\mathcal{G} = (\mathcal{V}, \mathcal{E})$ with a **node set** $\mathcal{V}$ and an **edge set** $\mathcal{E}$. One key property distinguishes the CPS graph from many other graph problems: in CPS, each individual **node** $i \in \mathcal{V}$ possesses a physical-world location $\boldsymbol{u}_i \in \mathbb{R}^2/\mathbb{R}^3$. Such physical location exists as an immutable attribute, and exerts a special constraint to CPS. Unlike pure cyber world graphs such as user-product graphs where the information can be instantaneously propagated from one node to

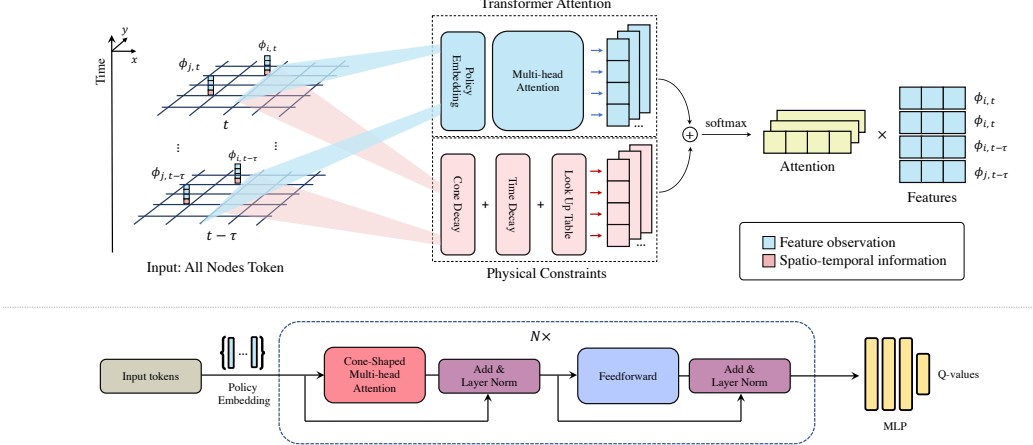

Figure 2: The DePT illustration of self-Attention layers (above) and the internal structure (below).

another, the information can can only propagate under a limited speed in the physical world, and therefore the interactions between nodes in CPS are governed by their physical distances.

We are interested in the control problem in CPS, where each node is an agent that constantly interacts with the environment as well as with each other. At every timestamp $t$, the agents observe the state $\boldsymbol{s}^{(t)}$ from state space $\mathcal{O}$, and take actions $\boldsymbol{a}^{(t)}$ from the action space $\mathcal{A}$. The system then get a reward $R^{(t+1)}$. The target of optimizing the CPS control is to come up with a collective optimal policy $\pi_{\mathcal{V}}^*$, under which the accumulated future reward is maximized:

$$\max_{\pi_{\mathcal{V}}} \sum_{k=0}^{\infty} \gamma^k R^{(t+k+1)} \tag{1}$$

where $\gamma \in [0,1]$ is the discount factor. In the discrete action space settings, the agents can be trained with Q-learning types of algorithms:

$$\min_{\boldsymbol{\theta}} \mathcal{L}(\boldsymbol{\theta}) = \mathbb{E}\left[ R^{(t+1)} + \gamma \max_{\boldsymbol{a}^{(t+1)} \in \mathcal{A}} Q(\boldsymbol{s}^{(t+1)}, \boldsymbol{a}^{(t+1)}; \boldsymbol{\theta}) - Q(\boldsymbol{s}^{(t)}, \boldsymbol{a}^{(t)}; \boldsymbol{\theta}) \right]^2, \tag{2}$$

where $\boldsymbol{\theta}$ is the parameters of the the learned Q-function $Q(\boldsymbol{s}, \boldsymbol{a}; \boldsymbol{\theta}) : \mathcal{O} \times \mathcal{A} \to \mathbb{R}$

## 4 DePT: Delayed Propagation Transformer

### 4.1 Preliminary: Transformers for Cyber-Physical Systems

We begin by introducing the transformer as the centralized agent to handle control problems in CPS. The transformer takes series of inputs, collected from all nodes spatially, and across the most recent $T_{max}$ timestamps temporally, making up the collected inputs as: $\mathbf{X} = \{\phi_{i,t-\tau}; i \in \mathcal{V}, \tau \in \{0, 1, \cdots, T_{max}\}\}$. Every $\phi_{i,t-\tau}$ in $\mathbf{X}$ is referred to as a **token**, which can be uniquely indexed via the ID of its corresponding physical **node** and time difference relative to the current timestamp $t$.

For each token $\phi_{i,t-\tau}$, in addition to feeding the transformer with the observed features (denoted as $\boldsymbol{f}_{i,t-\tau}$), we also make our transformer aware of the policy information via the *policy embedding*: we initialize a trainable embedding matrix $\boldsymbol{P} \in \mathbb{R}^{E \times |\mathcal{A}|}$, and for each action $\boldsymbol{a}_{i,t-\tau} \in \mathbb{R}^{|\mathcal{A}|}$ taken by node $i$ at time $t - \tau$, we index out the corresponding embedding $\boldsymbol{p}_{i,t-\tau} = \boldsymbol{P}[:, \boldsymbol{a}_{i,t-\tau}]$, and concatenate $\boldsymbol{p}_{i,t-\tau}$ with the features $\boldsymbol{f}_{i,t-\tau}$, and use $[\boldsymbol{f}_{i,t-\tau} \| \boldsymbol{p}_{i,t-\tau}]$ as the input.

For each pair of tokens $\phi_{i,t-\tau}$ and $\phi_{j,t-\rho}$, where $i, j \in \mathcal{V}$ and $0 \leq \tau, \rho \leq T_{max}$, the traditional design of transformer will compute the pre-softmax attention via:

$$a^{(T)}\left(\phi_{i,t-\tau}, \phi_{j,t-\rho}\right) = \phi_{i,t-\tau}^{\top} \boldsymbol{W}_Q^{\top} \boldsymbol{W}_K \phi_{j,t-\rho} \tag{3}$$

where $\boldsymbol{W}_K$ and $\boldsymbol{W}_Q$ are weight matrices to compute the key and query components, respectively. For every token $\phi_{i,t-\tau}^{(l)}$ at the $l$-th layer, its output from an attention layer is

$$\phi_{i,t-\tau}' = \sum_{j \in \mathcal{V}, \rho \in [T_{max}]} e\left(\phi_{i,t-\tau}, \phi_{j,t-\rho}\right) \boldsymbol{W}_V \phi_{j,t-\rho} \tag{4}$$

where $e\left(\phi_{i,t-\tau}, \phi_{j,t-\rho}\right) = \mathrm{softmax}_{j \in \mathcal{V}, \rho \in [T_{max}]}\left(a^{(f)}\left(\phi_{i,t-\tau}, \phi_{j,t-\rho}\right)/\Gamma\right)$, and $\Gamma$ is the temperature factor, the weight matrix $\boldsymbol{W}_V$ maps the input to the value component, and $e(\cdot, \cdot)$ denotes the pairwise post-softmax attention. In multi-head attention, each attention head will have an output $\phi'^{(k)}, k = 1, \cdots, N$, where $N$ is the number of attention heads. The computed attention heads are concatenated together and remapped to the output dimension via a linear block.

Following the attention layer, we append a layer normalization module with a skip connection. Subsequently, the output will be fed into a Feed-Forward Network (FFN) followed by a skip-connected layer with normalization. Multi-head attention, normalization, and FFN constitute a single encoding layer. We stack such layers building a $L$-layer transformer. At the output layer, we only read out the feature embedding of the nodes at the current time $t$, i.e., $\phi_{i,t-\tau}, \forall i \in \mathcal{V}$. Afterward, we employ another fully connected layer to map the feature embedding to the pre-action q-values. The architecture is illustrated in Fig. 2.

### 4.2 Enforcing Spatial-temporal Constraints

As illustrated in Fig.1, the spatial-temporal token pairs that have strong connections are more likely to be located around the cone-shaped area. We argue that in order to capture the most relevant spatial-temporal attention, the transformer agent operating in such settings should have their attention to follow the distribution of a cone-shaped prior.

To enforce such prior attention distribution, we train a new function $a^{(D)}(\phi, \phi')$ ($D$ for DePT), in addition to the Traditional transformer attention $a^{(T)}$ in Eq.3. These two functions are added up to obtain the pre-softmax attention value. Further, $a^{(D)}(\phi, \phi')$ consists of three components: the learned pair-relation Look-Up-Table embedding $\lambda^{(attn)}[i, j]$ (**LUT**), the cone shaped correlation decay function $\gamma(\epsilon_{j,\rho;i,\tau})$ (**ConeDecay**), the temporal correlation decay function $\sigma(\tau - \rho)$ (**TimeDecay**).

**Look-Up-Table embedding (LUT).** To capture the location-associated patterns, we enable our DePT to learn a scalar attention value $\lambda^{(attn)}[i, j]$ for every pair of nodes $(i, j)$ (tokens associated to the same nodes but with different timestamp share the same embedding). Such LUT is learned independently for every attention head in the original transformer.

**ConeDecay.** In the physical part of CPS, both the effect of nodes' actions and the passage of features can be modeled as *information flow* that are constantly exchanging/propagating between nodes. Think of a information flow with speed $v$ that originates from node $j$ at time $t - \rho$, and heads for node $i$ at time $t - \tau$. This flow will induce a causal connection in the physical world between two associated tokens $\phi_{i,t-\tau}$ and $\phi_{j,t-\rho}$. It is straightforward to define the causal deviation as follows:

$$\epsilon_{j,\rho;i,\tau}(\phi_{j,t-\rho} \rightarrow \phi_{i,t-\tau}) = (\tau - \rho)v - \|\boldsymbol{u}_i - \boldsymbol{u}_j\| \tag{5}$$

Under this definition, $\epsilon_{j,\rho;i,\tau} < 0$ and $\epsilon_{j,\rho;i,\tau} > 0$ corresponds to the expected "past" and "future" for the incident "$\phi_{j,t-\rho} \rightarrow \phi_{i,t-\tau}$ is exactly reachable by flow with speed $v$". We use the function $\gamma(\epsilon_{j,\rho;i,\tau}; W_\gamma)$ parameterized by $W_\gamma$ to penalize those "unreachable" token pairs, hence $\gamma(\cdot)$ is expected to obtain its global maximum at $\epsilon_{j,\rho;i,\tau} = 0$, and is a bi-directional decreasing function. It's gradient $\left|\frac{\mathrm{d}\gamma(\epsilon_{j,\rho;i,\tau_{ij}})}{\mathrm{d}\epsilon_{j,\rho;i,\tau_{ij}}}\right|$ depicts how fast the attention should drop and how "wide" the attention should spread, for the token $\phi_{j,t-\rho}$ that deviates from the cone of $\phi_{i,t-\tau}$. For the estimation of the speed, $v$ is the average of three learnable functions: the origination/destination speed, and a speed LUT (different from the above $\lambda^{(attn)}$): $\hat{v} = \frac{1}{3}\left(\nu^{(o)}(\phi_{j,t-\rho}'; W_{\nu^{(o)}}) + \nu^{(d)}(\phi_{i,t-\tau}'; W_{\nu^{(d)}}) + \lambda^{(v)}[i, j]\right)$

**TimeDecay.** Information from more distant past should be payed with less attention. To fit such time dissipation effect, given two tokens $\phi_{i,t-\tau}, \phi_{j,t-\rho}$, we let our DePT learn an additive function $\sigma(\tau - \rho; W_\sigma)$ parameterized by $W_\sigma$ for the pairwise attention.

**Overall Attention Form.** The ConeDecay, TimeDecay, and LUT are all learned separately for every transformer encoder block and every attention head. To further guide the capturing of desired attention as well as accelerate the training/inference in two-fold speed, we mask out the attention between token pairs $(\phi_{i,t-\tau}, \phi_{j,t-\rho})$ whenever $\rho > \tau$, that is, not letting node $n_i$ pay attention to node $n_j$ when $n_j$ is in the future timestamp of $n_i$ and have no chance to pass information to $n_i$. Distilling the essence up to this point, the architectures of the proposed DePT encoder blocks is shown in Fig. 2, and the pre-softmax attention between token pair $(\phi_{i,t-\tau}, \phi_{j,t-\rho})$ is computed via:

$$Attn(\phi_{i,t-\tau}, \phi_{j,t-\rho}) = \begin{cases} -\infty & \text{, if } \rho > \tau \\ \gamma(\epsilon_{j,\rho;i,\tau}) + \sigma(\tau - \rho) + \lambda^{(attn)}[i,j] \\ \quad + \phi_{i,t-\tau}^{\mathrm{T}} W_Q^{\mathrm{T}} W_K \phi_{j,t-\rho} & \text{, Otherwise} \end{cases} \quad (6)$$

### 4.3 Training DePT Faster: Prior Pre-fitting and Imitation Learning

Same as all other centralized control methods, transformers are not immune to slow convergence issues during RL. To leverage the favorable attention prior of DePT, We propose to pre-train the ConeDecay, TimeDecay, LUT components of DePT before interacting with the environments. In addition to pre-training, we also adopt imitation learning in the early stage of reinforcement learning.

**Pre-fitting the Priors.** There are six learnable components in every group of priors: the ConeDecay function $\gamma(\cdot)$, three speed estimators to compute $\epsilon_{j,\rho;i,\tau}$ for the ConeDecay: $\nu^{(o)}(\cdot), \nu^{(d)}(\cdot), \lambda^{(v)}[\cdot, \cdot]$, the TimeDecay function $\sigma(\cdot)$, and the attention LUT $\lambda^{(attn)}[\cdot, \cdot]$. Above them, the attention LUT and the speed LUT are matrices and can be initialized as $M_0$ and $M_{\bar{v}}$ respectively, where $M_x(x = 0, \bar{v})$ stands for a random matrix with all entries i.i.d. and have mean value $x$, and $\bar{v}$ is the average flow speed according to the statistics of the system. $\gamma(\cdot)$ and $\sigma(\cdot)$ are univariate functions, and can be pre-trained to fit certain desired analytical functions. In practice, we use $y = -kx^2$ to pre-fit $\gamma(\cdot)$ and $\sigma(\cdot)$, where $k$ is the normalization factor depending on the variance of input features. The two last functions $\nu^{(o)}(\cdot), \nu^{(d)}(\cdot)$, though multi-variate, can still be easily pre-fitted to labels sampled from $\mathcal{N}(\bar{v}, 0.1)$, and also lead to good initializations. All pre-fittings/initializations involve no costly tensor computation, and can therefore be accomplished within minutes by CPU. Such pre-fittings/initializations can largely boost the convergence of DePT, as to be shown in section 6.1.

**Warm-up with Imitation Learning.** Since the decentralized controller is typically easier to converge than the centralized method, we adopt the imitation learning(IL) [51] to warm-up RL. Before RL, a decentralized baseline controller is first trained and acts as the teacher model. During the IL stage, the actions are imitated by DePT and evaluated by the teacher model. During the subsequent training, DePT takes actions and evaluates the Q-values by its own interaction.

## 5 Experiments and Discussions

We use the network-wide urban traffic signal control (TSC) problem under connected and autonomous vehicles (CAVs) environment as a CPS control example to illustrate the performance of DePT. CAVs have great promises to help improve the performance of traffic signal control system, since they can provide traffic information (i.e., vehicle position, speed, acceleration, etc.) that are crucial to determining the signal phases and timings. The network-wide TSC under CAV is a multi-agent control problem where distributed control methods will encounter the non-MDP property for a single agent. In contrast, centralized control methods usually suffer from inefficient learning and slow convergence. In this section, we verify the ability of DePT under this challenging setting. We first define the TSC problem in Section 5.1 and introduce the dataset and simulation tools in Section 5.2. Then, we show the simulation results and discuss them in Section 5.3.

### 5.1 Network-wide Urban TSC problem

In the TSC setting, the graph is the urban traffic network, where the nodes/agents represent intersections and edges indicate the connecting roads between intersections. The viable action at each node is the traffic light signal to control the traffic flow. For a specific node, we take the surrounded traffic information provided by CAVs as the state, and use the signal phases as actions. Specifically, assume there is $k_i$ incoming lanes for intersection $i$, we define the state of this intersection as:

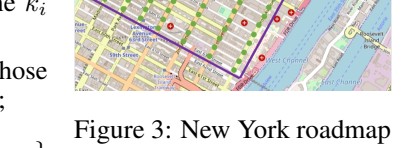

- $curTime \in \mathbb{R}$: current time in simulation;
- $numIn \in \mathbb{N}^{k_i}$: the number of CAVs on each of the $k_i$ lanes;
- $numQue \in \mathbb{N}^{k_i}$: the number of stopped CAVs (i.e., those CAVs with speeds $< 0.1m/s$) on each of the $k_i$ lanes;

Figure 3: New York roadmap considered in the experiments.

For intersection $i$, we define the action space as $a_i = \{1, 2, ...n_i\}$ where $n_i$ is the total number of phases for intersection $i$. As an example, consider an intersection $i$ that has two phases (i.e., $n_i = 2$): {West→East, East→West} and {South→North, North→South}, we define $a_i = 1$ as West→East/East→West green and the other conflict phase red, $a_i = 2$ as South→North/North→South green and the other conflict phase red.

## 5.2 Dataset

We use Cityflow [52] as the simulation platform due to the high computation efficiency. To evaluate DePT, we run experiments on both synthetic and real urban traffic networks built by [41]. The data is for scientific research usages, which is open-source and does not contain offensive content.

For the synthetic traffic networks, we use a $6 \times 6$ synthetic grid network with 36 agents taking actions every 10 seconds. There are 12 movements associated with every intersection (i.e., West→East, West→North, West→South, East→West, East→South, East→North, South→North, South→West, South→East, North→South, North→East, North→West). We design four phases for each intersection: {West→East, West→South, East→West, East→North}, {West→North, East→South}, {South→North, South→East, North→South, North→West} and {South→West, North→East}. There are 12 incoming lanes for each intersection, and the length of each lane is $300m$. Similar to [41], we test two flow settings for this network: bi-directional (noted as Grid-Bi) and uni-directional (noted as Grid-Uni). In Grid-Bi, we generate 300 vehicles/lane/hour for West→East and East→West movements and 90 vehicles/lane/hour for South→North and North→South movements. In Grid-Uni, we generate the same flows, but only for West→East and North→South movements.

For the real-world scenarios, we use the upper east side of Manhattan, New York, with $28 \times 7$ active agents as shown in Figure 3. The road network is imported from OpenStreetMap, and the flow is extracted by taxi data. For the evaluation metrics, we use the average travel time (noted as AvgTT, in seconds) of all vehicles, and the average queue length (noted as AvgQue, in vehicles).

## 5.3 Experimental Settings and Results

The number of different timestamps to be fed into the DePT ($T_{max}$) is set to be 10 in our experiments. The DePT is trained for 200 rounds for each dataset, while the first 100 rounds are imitation learning and the second 100 rounds are independent training. During each round, we run simulations for 4 hours and record state/action pairs, then we train DePT for 100 epochs with Double-DQN using the newly generated data. DePT is compared with three typical network-wide urban TSC methods:

- **Fixed-time** [53]: A classical signal planning method that uses daily vehicle volumes to calculate pre-timed signal plans based on operation manuals. The signal plans will not change once they are generated.
- **Max-pressure** [54]: A network-wide signal control method that dynamically generates greedy signal plans by maximizing the "pressure" (calculated from queue length) of upstream and downstream roads.

| Dataset | Grid-Bi | | Grid-Uni | | NewYork | |
|---|---|---|---|---|---|---|
| | AvgQue | AvgTT | AvgQue | AvgTT | AvgQue | AvgTT |
| Fixed-time[53] | 0.11 | 225.62 | 0.06 | 225.62 | 1.72 | 1957.35 |
| Max-pressure[54] | 0.09 | 208.13 | 0.04 | 198.93 | 1.75 | 1628.64 |
| GTN[29] | 0.06 | 204.34 | 0.05 | 192.44 | 1.73 | 1688.30 |
| Co-light[41] | **0.05** | 191.66 | **0.04** | 188.23 | 1.69 | 1605.56 |
| DePT | **0.05** | **184.22** | **0.04** | **180.71** | **1.52** | **1555.47** |

Table 1: Evaluation results of DePT against others.

| Stages | Imitation Learning | | | | Independent Training | | | |
|---|---|---|---|---|---|---|---|---|
| Rounds | 0 | 30 | 60 | 90 | 100 | 130 | 160 | 190 |
| DePT | 1143.1 | **244.3** | **211.6** | **202.4** | **199.6** | **188.2** | **185.0** | **184.9** |
| no-pre-fit | **997.2** | 782.1 | 243.8 | 218.9 | 216.2 | 192.4 | 190.1 | 189.2 |
| no-ConeDecay | 1042.6 | 616.4 | 317.3 | 227.7 | 225.2 | 240.3 | 285.4 | 269.3 |
| TTE | 1215.4 | 1418.9 | 955.8 | 326.7 | 299.5 | 265.9 | 297.1 | 313.7 |

Table 2: Ablation study results: AvgTT on Grid-Bi. TTE refer to the traditional transformer encoder.

- **Co-light** [41]: A state-of-art multi-agent RL-based network-wide signal control method that uses graph attentional networks to incorporate the temporal and spatial impacts among different intersections.

The teacher model we use during the IL stage is the Co-light model. The results for all models are shown in Table 1. As can be observed, DePT achieves the best performance among the four models, and it especially outperforms its teacher model Co-light.

## 6 Ablation Studies and Model Interpretations

### 6.1 Ablation Studies

We report two ablation studies of DePT: the effect of the prior pre-fitting, and the existence of newly proposed priors. Since the convergence rate is prohibitively slow if no imitation learning, all experiments are still executed with imitation learning. The AvgTT results on Grid-Bi dataset are shown in Table 2. In the table, the "DePT" corresponds to the proposed DePT that has its priors pre-fitted, where the "no-pre-fit" corresponds to the DePT without pre-fitting prior, the "no-ConeDecay" is the DePT without ConeDecay component but still pre-fitted other prior, and the TTE corresponds to the traditional transformer encoder without any prior. As can be observed, the "DePT" achieves the fastest convergence speed as well as the best performance, where the "no-pre-fit" model can also slowly converge to satisfactory performance. However, when the ConeDecay component is removed, both the "no-ConeeDecay" and the TTE model fail to converge properly.

### 6.2 Model Visualizations

The attention patterns are visualized in figure 4. Subfigure (a) is a visualization of the urban grid roadmap, while subfigures (b)-(g) are sampled attention matrices at the 30-th timestamp, coming from the 1st attention head at the last transformer encoder block. In all subfigures in (b)-(g), we display the attention as 4 parts: ❶ the ConeDecay prior $\gamma(\cdot)$, ❷ the TimeDecay prior $\sigma(\cdot)$ plus the attention LUT $\lambda^{(attn)}[\cdot, \cdot]$, ❸ the traditional attention $a^{(T)}(\cdot)$ as a residual component, and ❹ the overall attention, which is the summation of the previous three components.

Denote the size of the grid as $h \times w$, then the token number is $N_{token} = T_{max} \times h \times w$. The subfigures (d)(e)(f)(g) are the $N_{token} \times N_{token}$ attention matrix $\boldsymbol{A}$ itself, coming from the Grid-Bi dataset. The attention matrix in (b)(c) come from the New York dataset, and are sampled in the following way: we pick the certer node $n_0$ of the $7 \times 28$ grid of New York, then select all 28 nodes that are in the same vertical line as $n_0$. Then we extract the attention from $\boldsymbol{A}$ that correspond to the attention $n_0$ paid to all these 28 nodes in the past $T_{max}$ timestamps, forming a sub-matrix of $\boldsymbol{A}$ of shape $T_{max} \times h$ (reshaped from the union of multiple discontinuous segments in certain rows of $\boldsymbol{A}$).

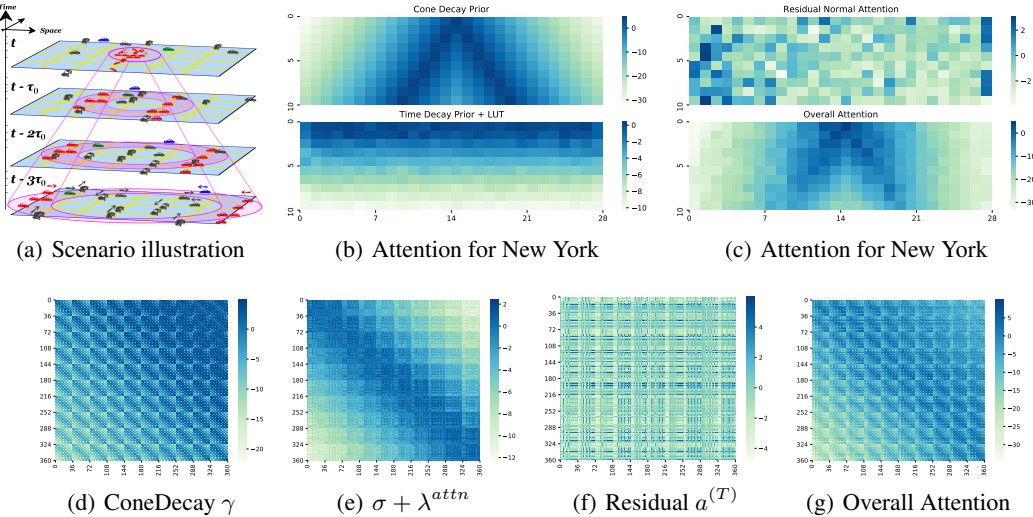

(a) Scenario illustration     (b) Attention for New York     (c) Attention for New York

(d) ConeDecay $\gamma$     (e) $\sigma + \lambda^{attn}$     (f) Residual $a^{(T)}$     (g) Overall Attention

Figure 4: Visualized Attention Distributions

As can be seen from the figures, the overall learned attention pattern displays the decay effect both along the cone-deviating direction and along the time axis, which is the same as it is designed for. Observing the cone attention in subfigure (b) (the above one), the learned cause decay effect shows asymmetry: the outwards direction (relative future) decays faster than the inwards direction (relative past), which is in line with the underlying Poisson distribution for the arrival of traffic flows. Note that such attention patterns rely on the instant dynamic features, and hence can vary across time, center tokens considered, and across different encoder blocks. Such observations demonstrate the interpretability of our DePT.

# 7 Conclusions

In this paper, we targeted to address several major challenges in multi-agent CPS control: the large state/action space for centralized methods, and the tension between localized observations and global control targets for decentralized methods. We proposed a new transformer-based model for feature learning and control in CPS – the Delayed Propagation Transformer (DePT). DePT induces physically-inspired attention priors into the transformer, which help boost the transformer's ability to capture the desired attention patterns. We choose one of the most complicated open-word system – traffic network to test the potential of performance of DePT on both synthetic and real open-source datasets, and our result shows that DePT not only converges faster and better than the traditional transformer but also outperforms other expert neural networks for traffic network control.

We see several technical and practical impacts of DePT. As a successful global-view controller, DePT offers a centralized control scheme that has the potential to solve the control problem under limited sensing. On the other hand, DePT is designed for general CPS under the constraint of limited physical propagation speed, hence is ready to plug-in for many other CPS problems, such as electrical power grids, industrial control systems, and the e-commerce delivery systems.

# Acknowledgement

Z.W. is supported by a US Army Research Office Young Investigator Award (W911NF2010240), and a J. P. Morgan Faculty Research Award.

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
