# OpenReview forum: "Delayed Propagation Transformer: A Universal Computation Engine towards Practical Control in Cyber-Physical Systems"
_NeurIPS.cc/2021/Conference — NeurIPS 2021 Poster_

### Official Review · Reviewer_pDRN · 2021-07-12

**Rating:** 8
**Confidence:** 4

**Summary:**

This paper proposed a new transformer for feature learning and control in CPS, called Delayed Propagation Transformer (DePT). DePT induces cone-shaped spatial-temporal attention priors into the transformer, which help boost the transformer’s ability to capture the desired attention pattern as ruled by immutable physical constraints. The authors demonstrated their new model on a traffic signal control application.

**Limitations And Societal Impact:**

No significant risk or concern

**Main Review:**

Strength
-	This is the first paper leveraging transformer to multi-agent system control as far as I know.
-	To model the information propagation in physical systems, the authors pointed out that the correlation pattern in space-time follows a neighboring distribution of the cone-shaped distribution, which is set as an important prior for the transformer attention.
-	The model seems “universally” applicable to model CPS dynamics, due to being free of inductive bias while being physics aware.
-	The authors conduct thorough experiments on both synthetic and real urban traffic networks and observe new state-of-the-art results. The overall learned attention pattern also displays the desired decay effect.
-	The paper is clear and easy to read. The releases of codes and data are promised.

Weakness
-	I am not convinced that transformer free of locality-bias is indeed the best option. In fact, due to limited speed of information propagation, the neighborhood agents should naturally have more impacts on each other, compared to far away nodes. I hope the authors to explain more why transformer’s no-locality won’t make a concern here.
-	Due to the above, I feel graph networks seem to capture this better than the too-free transformer, and their lack of global context/ the “over-squashing” might be mitigated by adding non-local blocks (e.g., check “Non-Local Graph Neural Networks” or several other works proposing “global attention” for GNNs).
-	The authors also claimed “traditional GNNs” cannot handle direction-feature coupling: that is not true. See a latest work “MagNet: A Neural Network for Directed Graphs” and I am sure there were more prior arts. Authors are asked to consider whether those directional GNNs can possibly suit their task well too.
-	Transform is introduced as a centralized agent. Its computational overhead can become formidable when the network gets larger. Authors shall discuss how they prepare to address the scalability bottleneck.


**Time Spent Reviewing:**

4h

---

> ### Author Response · Authors · 2021-08-10
> **Response to Reviewer pDRN**
>
>
> We genuinely appreciate your valuable suggestions and raised concerns. We tried our utmost to address all raised questions as follows.
>
> **Q1: Why transformer's no-locality won't make a concern, considering the neighborhood agents naturally have more impacts on each other.**
>
> **Reply:**
>
> Thanks for pointing out our exact claim: transformers' non-locality is not always good under CPS' delayed propagation. To be precise, "transformers" do not have locality inductive bias, however, DePT does. Transformers are fully global-view methods, and GNNs are fully local-view methods. On the other hand, DePT has both a global view and a local prior: it brings advantages from two extreme sides.
>
> Being a transformer, DePT gains the flexibility of simultaneously handling spatial and temporal data as unified tokens, and gains from the transformer's “decentralized processing” (through token-wise feedforward processing) and “centralized information aggregation” (through global attention). In the meantime, it inherits transformers' non-locality, which are partially credited to achieve better performance, but are not always compliant to the underlying physics.
>
> To make transformers better physics-aware, DePT moderately adds the locality prior back, using the proposed ConeDecay and TimeDecay. In Particular, the cone decay can be interpreted as a space-time joint locality constraint: "the further two incidents deviate from the propagation cone, the weaker the attention".
>
> Your concern is exactly our motivation for adding these special priors to transformer attention blocks, making DePT different from traditional transformers and more physics-aware.
>
> **Q2: Traditional GNNs are able to handle direction-feature coupling.**
>
> **Reply:** We are appreciated for this comment. We will rectify this statement in the final version and will cite MagNet in the final version.
>
> **Q3: As a centralized agent, the computational overhead can become formidable when the network gets larger.**
>
> **Reply:**
>
> We agree with you that the complexity will increase as the number of agents increase, but as of the common scales considered in nowadays' state-of-the-art researches, the scalability is far from being a bottleneck to DePT. Specifically, the maximal number of intersections considered are: 24 in Intellilight [34],  16 in Presslight [35], 196 in Colight [37], 16 in Metalight [39], 11 in AttendLight [40]. DePT handles 196 agents (intersections) in the NewYork dataset.
>
> On the other hand, we would like to comment that compared to traditional centralized methods, DePT is already reducing the size of action space from exponentially growing to constant, and reducing the size of state space from linearly growing to constant, w.r.t. the agent quantity. Consider a CPS with $M$ agents. Consider the number of actions is $Q$ for each agent, and the dimensionality of state space is $d$ for each agent. For traditional centralized agents, the total number of output actions are $Q^M$, requiring the output layer of the "actor" to have $Q^M$ number of logits, and the entire state space has dimensionality $M*d$ (if without special graph pooling techniques). For DePT, however, thanks to its "decentralized processing" (not to be confused with “decentralized methods”: DePT belongs to the “centralized” method, but the feedforward processing is “decentralized”) and shared feedforward network, the output layer only have $Q$ logits (constant w.r.t. $M$), and the feedforward network only needs to deal with $d$-dimensional constant state space. The way DePT qualifies for a "centralized method" is that DePT has global information routing modules (the attention heads), however these modules have nothing to do with increasing the state space and action space.

---

> > ### Comment · Reviewer_pDRN · 2021-08-18
> > **My concerns are addressed**
> >
> > The rebuttal addresses my concerns and also seems to answer the questions of others carefully. I support to accept it. Thanks.

---

### Official Review · Reviewer_BFpw · 2021-07-12

**Rating:** 7
**Confidence:** 4

**Summary:**

The authors used transformer to model the spatial-temporal dynamics in a real-world system with information propagation delay, which is claimed to be specialized at global modeling. The transformer design has taken the physical constraints into consideration. It shows promise in learning to control multi-agent systems.

**Ethical Concerns:**

Not found

**Limitations And Societal Impact:**

No significant social risk

**Main Review:**

The main novelty of this paper is to bake in physics prior in transformers to model CPS. Previous works utilized transformer to model networks or graphs, but usually from non-physical systems and without a notion of information propagation latency. By taking the important physical delay constraint into account, the new transformer has essentially restricted a physically realizable subspace of solutions, which the authors argue to be the key for modeling CPS.

To formulate the constraint, the authors designed a cone-shaped temporal-spatial prior, as the feature propagation in physical world process naturally forms the cone shape in the time-space. That prior guide the design of a multi-level heterogeneous attention mechanism.

The proposed controller achieved the state-of-the-art performance in the traffic signal control task, in a large-scale network with heterogeneous node and edge types. Specifically, the new transformer approach outperformed the state-of-the-art Co-light method, that uses graph attentional networks to model the temporal-spatial space.

Overall, this is a solid paper tackling a worthy problem. The novelty is clear. The writing is also good quality.

Questions & Suggestions:

(1)	Transformer belongs to the centralized methods and can be hard to converge due to the huge state and action spaces. Although the authors presented two empirical tweaks that can alleviate training difficulty, I double the transformer approach may still have difficulty to scaling up to large amounts of agents in a real city (currently two experiments had only 36 and 196 agents). On the other hand, graph NNs may scale up better due to its locality-based update and aggregations.

(2)	Graph NNs also can apply to directional graphs too, e.g., “Directed Graph Convolutional Network”. Therefore, line 43-49 claims are inaccurate.

(3)	I think “universal” is too high in tone since the authors only demonstrated one application of traffic signal control. Similarly, the authors should avoid many exaggerative yet imprecise/unnecessary buzzwords, such as ““light cone/ Special Relativity”, or “Causality”.

**Time Spent Reviewing:**

4

---

> ### Author Response · Authors · 2021-08-10
> **Response to Reviewer BFpw**
>
>
> We genuinely appreciate your valuable suggestions to strengthen our paper. We tried our best to address all raised questions.
>
> **Q1: Transformer belongs to the centralized methods, hence can be hard to scale to large number agents in a real city due to the huge state and action spaces.**
>
> **Reply:**
>
> Thanks for raising this concern, but we would like to comment on your potential misunderstanding. Although DePT belongs to the centralized methods, it is free of the "large action spaces" issue that bothers traditional centralized methods, thanks to its "decentralized processing" (not to be confused with “decentralized methods”: DePT is a “centralized method” due to global attention, but uses “decentralized processing” due to token-wise feedforward network structure). Compared to traditional “centralized methods”, DePT reduces the action space from "exponentially growing w.r.t. agent quantity" to "constant w.r.t. agent quantity", and reduces the state space from "sub-linearly growing w.r.t. agent quantity" to "constant w.r.t. agent quantity".
>
>
> Prior to this work, existing "centralized methods" use "centralized processing": if the number of agents is $M$ and the number of actions is $Q$ for each agent, then at every single state, the total number of actions are $Q^M$. For centralized methods prior to DePT, this requires that the output layer of the "actor" should have $Q^M$ number of logits, which causes great instability in training, due to the apparent difficulty to generate enough data to fit all logits, when $M$ is large. DePT, on the other hand, benefits from the "decentralized processing" in that its output layer only has $Q$ logits, and each agent only selects one action out of the $Q$ logits. Meanwhile, since the feedforward network is shared across all agents, it is easy to fit all $Q$ logits, eliminating the above instability. Similar case apply to the state space: if each agent has a $d$-dimensional state space, the DePT agent will only need to process $d$-dimensional state space in a decentralized way, while traditional centralized methods need a centralized network to process $M*d$-dimensional state space, or as a mitigation, they could pre-reduce it to a lower dimension via graph-pooling based approaches.
>
> In summary, DePT enjoys the benefit of being a "centralized method" due to its global information routing in the attention layers, while also being free of the huge action space issue thanks to its "decentralized processing".
>
> We promise to add the above discussions to the final version upon acceptance.
>
>
> **Q2: Graph NNs can also apply to feature-direction coupled cases.**
>
> **Reply:** We appreciate this comment. We will rectify this statement and cite directed GCN paper in the final version.
>
> **Q3: The authors should avoid many exaggerative yet imprecise/unnecessary buzzwords.**
>
> **Reply:** We genuinely appreciate this advice. We will resolve imprecise expressions in the final version.

---

### Official Review · Reviewer_2aDi · 2021-07-15

**Rating:** 7
**Confidence:** 4

**Summary:**

Cyber-Physical Systems (CPS), which is used to control and monitor the health of multiple agents and systems, and predict their future health, has a major challenge in effectively implementing multi-agent control. In CPS, an ideal MDP cannot be established due to incomplete coverage/placement of the sensor and limitations of current sensing technology. The author’s purpose is a new transformer-based Delayed Propagation Transformer (DePT) model that can specialize in global modeling of CPS while considering the constraints of the physical world. The transformer-based model gets rid of the agent’s inherent locality restrictions, and learns global views of correlations across the whole system. The author customized a multi-level heterogeneous attention mechanism that has the causality cone prior backed into its design. With such design, the attention between input tokens now has a physical interpretation, hence guiding the controller to learn more effectively in exploring the collaborative strategy across agents. Taking advantage of these strengths, we plan to build a DePT-based CPS controller and use it in a traffic signal control system.

**Limitations And Societal Impact:**

The simulator is a real product used in the real world or a fake product that has a similar feel to it. Either way, since they are ‘real things’ that exist under the physical environment, physical limitations and operation and maintenance will inevitably require considerable time and cost. The simulators that overcome these physical limitations do not exist yet, so there is always an error between the real world and the virtual world. In order to overcome this limitation, this paper needs to be proven by operating it in a real environment.

**Main Review:**

In general, the techniques of CPS and the problem of multi-agent control are well known. Edges in the  CPS  system graph have physical orientations, and these orientations are inherently coupled with node/edge functions and can be translated into completely different meanings. For this reason, the representative problem of CPS control is the difficulty of training due to the physically limited space and noisy CPS data. The authors of this paper have a good understanding of the problems of the conventional transformer model and tried to solve the problem using a unique model called the Delayed Propagation Transformer (DePT). The authors have a good understanding of the problems with existing technologies and have technically described the proposed techniques to solve them in a technically friendly manner. However, it would have been better if the core difference between the existing Transformer model and the DePT model was clearly revealed. With a clear theoretical/experimental approach, the authors leveraged existing data to achieve better results than before. In addition, everything in the paper is well laid out and provides the reader with relevant information.

**Time Spent Reviewing:**

3

---

> ### Author Response · Authors · 2021-08-10
> **Response to Reviewer 2aDi**
>
>
> We genuinely appreciate your suggestions, and write below our response to them.
>
> **Q1: It would have been better if the core difference between the existing Transformer model and the DePT model was clearly revealed.**
>
> **Reply:**
>
> Thanks for this comment. The core difference between existing transformers and DePT is the physical awareness. Current transformers are not physics-aware, due to the fact that the global view does not comply to the underlying neighborhood information exchange. DePT on the other hand, uses the ConeDecay and TimeDecay (in section 4.2) to enable physics-aware cone-shaped attention distribution, making it better suitable for CPS tasks.
>
> **Q2: The simulators that overcome physical limitations do not exist yet, hence in order to overcome the limitations, DePT needs to be proven by operating it in a real environment.**
>
> **Reply:**
>
> Thanks for raising this concern. In previous CPS control related studies, it is a common practice to use high-fidelity simulation platforms, such as [34] - [40]. On the other hand, the simulations on high-fidelity platforms have witnessed alignments to real world experiments [a-d]. We will also try to evaluate the proposed method to natural environments upon obtaining sufficient industrial support.
>
> [a] Besselink B, Turri V, Van De Hoef S H, et al. Cyber–physical control of road freight transport[J]. Proceedings of the IEEE, 2016, 104(5): 1128-1141.
>
> [b] Givehchi O, Landsdorf K, Simoens P, et al. Interoperability for industrial cyber-physical systems: An approach for legacy systems[J]. IEEE Transactions on Industrial Informatics, 2017, 13(6): 3370-3378.
>
> [c] Leitao P, Karnouskos S, Ribeiro L, et al. Smart agents in industrial cyber–physical systems[J]. Proceedings of the IEEE, 2016, 104(5): 1086-1101.
>
> [d] Almási P, Moni R, Gyires-Tóth B. Robust reinforcement learning-based autonomous driving agent for simulation and real world[J]. arXiv preprint arXiv:2009.11212, 2020.

---

### Official Review · Reviewer_qqzv · 2021-07-17

**Rating:** 6
**Confidence:** 4

**Summary:**

The paper presents a transformer-based centralized method to approach the multi-agent control problem in cyber-physical systems (CPS). They formulate the graph-based CPS control as a reinforcement learning problem, and adapt transformer to represent the Q-network by introducing spatial-temporal constraints in computing self-attention. The constraints include a cone-shaped prior for neighboring distribution, a location-associated embedding and time decay effect. To accelerate training of such centralized method, they also propose to retrain three constraints and adopt imitation learning for warm-up. They experiment on one of the CPS scenarios, traffic signal control, and the results show the effectiveness of the proposed method compared to the baselines.

**Limitations And Societal Impact:**

The authors address two specific issues regarding how to adopt transformers in approaching the CPS control problem. However, the general limitations brought by centralized methods are still remained to be solved, e.g., as stated in the introduction section, "it is hard for the centralized methods to converge due to the huge state and action spaces". It is unclear how the proposed method deals with huge state and action spaces.

In addition, the motivation of using transformer is kind of vague. It would be better to clearly state the benefits of using transformer in solving this problem. The citations of transformers in line 51 are all about NLP and CV domains. Why transformers are good at removing locality inductive bias and capturing global view in CPS, compared to existing GNN based approaches. Also the direction-feature coupling is not a big issue for GNN, as directed graph neural networks [b][c] have been proposed.

[b] Zhang, Xitong, et al. "MagNet: A Magnetic Neural Network for Directed Graphs." arXiv preprint arXiv:2102.11391 (2021).
[c] Thost, Veronika, and Jie Chen. "Directed acyclic graph neural networks." arXiv preprint arXiv:2101.07965 (2021).

**Main Review:**

**== Originality ==**
This is the first work to approach CPS control problem via transformer framework and the idea of incorporating the cone-shaped spatial-temporal prior into the self-attention is interesting and new to me. In particular, the authors do not directly transformer on the CPS scenario, but make some improvements of the model in consideration of direction-feature coupling issue in directed graphs and time decay in message propagation. For the training convergence speedup, the pertaining RL agent for warmup is a common technique for RL-based application besides CPS and hence this part is not new to me.

**== Quality ==**
It is appropriate to adopt transformer for the CPS control problem and the modification in terms of the issues in centralized CPS is reasonable. However, I have some concerns regarding the DePT training speedup and the experiment results are not convincing to me.

For the training speedup, it seems that the slow convergence is partially caused by the proposed spatial-temporal prior constraints introduced in calculating transformer attention, and that's why additional pre-fitting priors are required in Section 4.3. In other words, the proposed acceleration method is only specific to the model in this work, but not generalized to other centralized methods for CPS control. Therefore, the common difficulty of training centralized methods under huge state and action space still cannot be addressed, as stated in the introduction section.

For the experiment part, there are three major issues.
1. The model is only evaluated on one scenario of CPS (i.e., traffic signal control) and hence the generalizability of the method under other CPS domains is unknown. Even for the selected traffic signal control task, only one real-world dataset (NYC city) is adopted for evaluation. The results would be more convincing if more real-world datasets (e.g., other cities used in CoLight [36]) are experimented.
2. The selected baselines are relatively weak. For example, the recent work Attendlight [39] can be considered as a strong baseline to demonstrate the effectiveness of the proposed model. Besides, no decentralized method such as [35][36] is selected for comparison. Since the authors claim that the proposed model can better capture the global-view of CPS system compared to the decentralized methods, it would be better to show it in the experiment.
3. The experiment analysis on the proposed model is not thorough. Since the major technical contribution of this work is on the modification of transformer, evaluation of other transformers variants on the CPS control is also encouraged. For example, the GraphTransformer [a] may be suitable in the graph-based CPS problem. In addition, the ablation study in Table 2 also fails to consider other centralized methods in evaluating the training convergence. Since the imitation learning is a common technique for any RL-based models, so what if it is applied to CoLight to speed up training?

[a] Yun, Seongjun, et al. "Graph transformer networks." Advances in Neural Information Processing Systems 32 (2019): 11983-11993.

**== Clarify ==**
In general, the paper is well-structured and easy to follow in most of the parts. However, some details in problem formulation and model description should be clarified.
1. The definition of MDP in CPS is unclear. I believe the detail of traffic signal control problem is not familiarized in the NeurIPS community and hence it would be better to clearly define the MDP formulation, e.g., what is  the reward function.
2.  In section 4.1, what is the policy embedding P and what is notation E? Since Q learning is adopted to approach the RL problem, why policy is required in this case? Is the observed features f correspond to section 5.1?
3.  Although the source codes are provided, it is still encouraged to describe implementation detail of the proposed method in the paper.
4.  There are many typos and grammatical errors in the paper, which should be fixed in the final version.
  - line 32, "the" -> "The"
  - line 78, "have" -> "has"
  - line 80, "learned" -> "learn"
  - line 113, "model" -> "modeling"
  - line 142, "interact" -> "interacts"
  - line 157, "feed" -> "feeding"
  - line 185, "consists" -> "consists of"
  - line 221, "pre-train" -> "per-training"
  - line 294, "feed" -> "feeding"

**== Significance ==**
The paper targets a specific task of traffic signal control rather than a general CPS problem. Although some technical contribution of adapting transformers in the task is novel to me, the evaluation is somehow problematic and the results are not convincing. The work may be only interesting to a small group of audience in the community.

**Time Spent Reviewing:**

6

---

> ### Author Response · Authors · 2021-08-10
> **Response to Reviewer qqzv**
>
> We geneuinly appreciate your concerns, and tried our best to address them.
>
> **Q1: The general limitations of huge state/action space for centralized methods are still remained to be solved. It is unclear how the proposed method deals with huge state/action spaces.**
>
> **Reply:**
>
> Thanks for raising this concern. Compared to traditional centralized methods, the proposed DePT reduces the action space from "exponentially growing" to "constant" w.r.t. agent quantity, and reduces the state space from "sub-linearly growing" to "constant" w.r.t. agent quantity.
>
> DePT overcomes the huge state and action space issue thanks to its "decentralized processing" (not to be confused with “decentralized methods”: DePT is a “centralized method” due to global attention, but uses “decentralized processing” due to token-wise feedforward network structure).
>
> Prior to this work, existing "centralized methods" use "centralized processing": if the number of agents is $M$ and the number of actions is $Q$ for each agent, then at every single state, the total number of actions are $Q^M$. For centralized methods prior to DePT, this requires that the output layer of the "actor" should have $Q^M$ number of logits, which causes great instability in training, due to the apparent difficulty to generate enough data to fit all logits, when $M$ is large. DePT, on the other hand, benefits from the "decentralized processing" in that its output layer only has $Q$ logits, and each agent only selects one action out of the $Q$ logits. Meanwhile, since the feedforward network is shared across all agents, it is easy to fit all $Q$ logits, eliminating the above instability. For state space, if each agent possess $d$-dimensional state space, then both for centralized DePT and decentralized methods, the actual state space need to be processed is $d$-dimensional, while for traditional centralized methods, that dimensionality is $M*d$, or as one potential mitigation, they could pre-reduce it to a lower dimensionality using graph-pooling based approaches.
>
> In summary, DePT enjoys the benefit of being a "centralized method" due to its global information routing in the attention layers, while also being free of the huge action space issue thanks to its decentralized processing. We will add the above discussions to the final version.
>
>
>
>
> **Q2: The results would be more convincing if other cities in Colight are experimented and Attendlight or [35][36] or GraphTransformer are compared.**
>
> **Reply:**
>
> We appreciate your suggestion. We have added experiments on the two remaining real word Asian cities in the general response Q1. We used Colight as our main baseline since it is an open-sourced well-cited algorithm. AttendLight is another strong SOTA algorithm not open-sourced. By the rebuttal deadline, we were unable to reproduce AttendLight to reliably close performance to what was originally reported, due to this short time frame. We are now contacting AttendLight authors and look forward to their help, and we hope to report new results whenever the successful reproduction is achieved. We added GraphTransformer as another baseline in general response.
>
> **Q3: What if applying imitation learning to CoLight to speed up training?**
>
> **Reply:**
>
> Colight is already converging fast, and is the best performing algorithm besides DePT. We expect letting Colight imitate another worse baseline could fasten it but not boost its final performance. We additionally added ablation study of imitation for DePT in general response Q1.
>
> **Q4: It would be better to clearly define the reward function, the policy embedding P, notation E; why policy is required; is the features f correspond to section 5.1?**
>
> **Reply:**
>
> We define reward as the negative of the total queue length of each intersection: $R^{t+1} = -\sum_{i = 1}^{N} \sum_{j = 1}^{k_{i}} numQue_{i,j}$ where $N$ is the number of intersections, $k_{i}$ is the number of lanes for intersection $i$, and $numQue_{i,j}$ is the queue length of the $j$th lane of the $i$th intersection. Thanks for your suggestion. We will add this in the final version. $P$ is an $E$ by $|A|$ matrix, where $E$ is the dimension of the embedding, and $|A|$ is the number of actions for each node. The reason policy is required is that all previously available information should be encoded. Colight leverages the historical actions as part of state information in the adjacency connections, and DePT does so with policy embedding. Yes, the f exactly corresponds to lines 262-265. We will add the above in the final version.
>
> **Q5: The proposed acceleration method is only specific to the model in this work.**
>
> **Reply:**
>
> We believe this statement comes out of an incomplete understanding of our algorithm, and we don't agree with it. To apply our acceleration to graph-based methods, one can modify the neighborhood aggregation by adding the proposed ConeDecay/TimeDecay terms on top of the original ”sum aggregation”, then apply “pre-fit” to those priors. So long as there is any neighborhood aggregation in the model (as all GNNs do), our architecture plug-ins and acceleration methods can generalize to it.
>
> As for the influence of this work, we believe this work has stand-alone values by itself, and could bring broad impact beyond traffic signal control in the CPS domain. The traffic signal control problem is one of the most challenging problems and a well-received benchmark in CPS, which has captured a number of previous researches such as [34] - [40]. We agree that there remains room for future work to generalize to other CPS problems, however the methodology seems straightforward and can be shared by other CPS tasks as well. We have planned to extend this work to more CPS settings: control under the delayed formation reshaping propagation in UAV swarms using Microsoft Airsim; centralized control in large-scale power grids using Grid2Op from RTE-France. For any networked CPS problem with delayed information propagation, our method should fit. Considering the versatile applications of CPS, we believe our work is of broad interest in the community.
>
> **Q6: It would be better to clearly state why transformers are good at removing locality inductive bias compared to GNN approaches.**
>
> **Reply:**
>
> Thank you for this comment. To be precise, "transformers" do not have locality inductive bias, however, DePT does. Transformers are fully global-view methods, and GNNs are fully local-view methods. Contrastively, DePT has both a global view and a local prior: it brings advantages from two extreme sides.
>
> Being a transformer. DePT inherits the non-locality, which are partially credited to achieve better performance, but are not always compliant with the underlying physics. It then moderately adds the locality prior back with “ConeDecay”/etc: "the further two incidents deviate from the propagation cone, the weaker the attention".
>
> Compared to existing GNN based approaches, DePT is also one type of GNN -- a fully connected GNN. The proposed priors softly exert spatial-temporal locality to DePT, analogous to GNNs' spatial locality. It allows for moderate global info routing whenever needed, and mediates between the demand of handling non-local info and the local physical info exchange, which makes DePT physics-aware.
>
>
>
> **Q7: The feature-direction coupling is not a big issue for GNN.**
>
> **Reply:**
>
> We appreciate this comment. We will rectify the statements in lines 43-49, and will cite MagNet, and Directed Acyclic GNN papers.

---

> > ### Author Response · Authors · 2021-08-24
> > **Sincerely expecting further discussions from reviewer qqzv**
> >
> > Dear Reviewer qqzv:
> >
> > We thank reviewer qqzv’s time for the review, and we really hope to have a further discussion with reviewer qqzv to see if our response solves the concerns.
> >
> > We have supplemented discussions and explanations about your concerns in the rebuttal. For example, in response to the huge state/action spaces concern for centralized methods as DePT, we have demonstrated that DePT is able to reduce the space sizes from “exponentially growing/sublinearly growing” to constants. Meanwhile, regarding the suggestion to add more comparison experiments, we have added experiments on GraphTransformer as a new baseline, together with two new datasets and three ablation studies.
> >
> > We genuinely hope reviewer qqzv could kindly check our response and and make this work known by more researchers together. Thank you!
> >
> > Best wishes,
> > Authors

---

> > > ### Comment · Reviewer_qqzv · 2021-09-01
> > > **My concerns are addressed**
> > >
> > > The authors provided detail feedbacks that addressed my major concerns on the experiment part. The results are more convincing to me now. Considering these, I will raise my score to 6 and will be fine if this paper gets accepted.

---

> > > > ### Author Response · Authors · 2021-09-02
> > > > **Follow up**
> > > >
> > > > Dear reviewer qqzv:
> > > >
> > > > We deeply appreciate your time and acknowledgement to our previous responses!
> > > >
> > > > Thanks a lot and best wishes,
> > > >
> > > > Authors

---

### Official Review · Reviewer_SpfT · 2021-08-03

**Rating:** 5
**Confidence:** 4

**Summary:**

The paper proposes a novel centralized multi-agent control technique. This technique relies on a novel transformer architecture that has been designed to account for the various domain-specific constraints in the dynamic multi-agent control setting, namely the presence of time delays in signal propagation across the network of agents. The paper details the specifics of this novel architecture and performs experiments on the CityFlow traffic control simulation platform. Various ablation studies analyze the contributions of various parts of the architecture to the overall system performance.

**Limitations And Societal Impact:**

Although I do not think it is necessary, the authors may want to more explicitly mention current limitations of the architecture and plans for future work.

**Main Review:**

Originality: The main contribution of the paper is in the design of the transformer architecture. This is certainly well motivated by the application area in which the model is to be deployed. However, the contribution seems incremental, and in some ways it is similar to common masking schemes that are widely used to prevent "cheating" of standard transformer architectures during training. A more thorough literature review of masking techniques and priors that are used in transformer architectures would be helpful context.

Quality: Overall the claims are generally supported, although some further detail could do a great deal to substantiate the novelty and importance of the proposed architecture. For example, some further ablation studies could supplement the existing analysis. Namely, imitation learning is done for all experiments. It is therefore unclear how much marginal benefit is provided by the novel architecture components on top of the baseline of imitation learning. Furthermore, experimental comparison with other priors proposed in the literature could be useful as well. In this sense, the paper appears to be a strong start but is still a work in progress.

Clarity: Aside from a few scattered grammar issues and typos, the paper is written clearly and is easy to follow. The organization makes sense for the proposed contribution (i.e. the lack of theoretical guarantees or analysis is expected for this line work).

Significance: As mentioned above, it is unclear how novel the contribution is. I think the significance of the work will be in in its ability to outperform other methods in benchmark tasks. Currently, the experiments are on a bespoke dataset/simulation platform so it is hard to compare against other widespread techniques. Further experimentation and comparison with existing techniques will take this work further into showcasing the significance of its novel transformer architecture.


**Time Spent Reviewing:**

2 hours

---

> ### Author Response · Authors · 2021-08-10
> **Response to Reviewer SpfT**
>
> We genuinely appreciate your valuable suggestions to strengthen our paper. We tried our utmost to address all raised questions, including discussions about our architecture's limitations and potential mitigations.
>
> **Q1: The contribution seems incremental.**
>
> **Reply:**
>
> We believe the contribution of this paper is not incremental. As pointed out by reviewer "qqzv": "this is the first work to approach CPS control problem via transformer framework"; as by reviewer "2aDi": "The author's purpose is a new transformer-based DePT model that can specialize in global modeling of CPS while considering the constraints of the physical world"; as by reviewer "BFpw": "It shows promise in learning to control multi-agent systems"; and as by reviewer "pDRN": "This is the first paper leveraging transformer to multi-agent system control as far as I know". Same with the reviewers’ opinions, we believe such a deep learning control scheme can show huge impact on both theoretical and practical perspectives, especially for the partially MDP control problem.
>
> **Q2: The contribution is similar to common masking schemes to prevent "cheating".**
>
> **Reply:**
>
> Our proposed prior is not used to prevent cheating, instead, it encourages the finding of appropriate attention patterns in the “non-cheated” data under the predictive control framework. The proposed priors are fundamentally different from common masking schemes in the two following ways. First, the underlying design has different interpretations. In all previous CV/NLP transformer research, the data does not follow the joint spatial/temporal cone-shaped distribution; while in CPS domain, there is no previous research leveraging the transformer. Second, the implementations are different. DePT does not mask "future data" that otherwise "cheats" -- the future data are never fed into DePT. The priors are only applied to historical input data of DePT.
>
>
> **Q3: Literature reviews on masks are helpful.**
>
> **Reply:**
>
> Thanks for your suggestion on enriching our literature reviews. We will add the below literature reviews into the final version.
>
> [a] considers padding mask and look-ahead mask. In encoder, the padding mask ensures all longer sequences are truncated while shorter sequences are padded. In the decoder, the look-ahead mask restricts the self-attention layer only to attend to earlier positions in the output sequence.
>
> [b] generalizes SAN and FFN to mask attention network (MAN), and proposes Dynamic Mask Attention Network (DMAN) with a learnable mask matrix that masks other tokens that are not in the neighborhood of the target token for better local semantic modeling.
>
> [c] proposes a differentiable masking mechanism for dense video captioning. The captioning decoder employs a masking network to restrict its attention to the proposal event over the encoding feature.
>
> For localness modeling, [d] casts localness modeling as a learnable Gaussian bias, and incorporates this bias into the original attention distribution as a new attention network, which indicates the central and scope of the local region to be paid more attention.
>
> [e] extends the self-attention mechanism to efficiently consider representations of the relative positions or distances between sequence elements through adding a relative position embedding to the key vectors.
>
> [a] Vaswani et al., Attention Is All You Need
> [b] Mask Attention Networks: Rethinking and Strengthen Transformer
> [c] Zhou et al., End-to-End Dense Video Captioning with Masked Transformer
> [d] Yang et al., Modeling Localness for Self-Attention Networks
> [e] Shaw et al., Self-Attention with Relative Position Representations
>
> **Q4: It is unclear how much benefit come from the novel architecture on top of imitation learning; experimental comparisons with other priors could be useful as well.**
>
> **Reply:**
>
> We follow your suggestions to conduct a new series of ablation study on imitation learning, and the results are presented in the general response Q1. For comparisons with other priors, we've added the ablation study for the TimeDecay and STLUT, which we believe when unioned with Table.2 in the original submission, they form complete ablation series for all newly proposed components and training strategies.
>
>
> **Q5: It is difficult to compare the current datasets / techniques / simulation platform against others.**
>
> **Reply:**
>
> Thanks for raising this concern. For mitigating the "dataset" and the "techniques", we have conducted additional experiments on two other real world datasets, and another graph-based method (GraphTransformer). The results are put in the general response.
>
> For the "simulation platform", indeed, experiments on certain CPS task depend on the high-fidelity system level simulation platform specifically designed for that task. On one hand, traffic signal control is one of the most challenging problem and a well-received benchmark in CPS, which has captured a number of previous researches, such as Intellilight [34], Presslight [35], Toward a thousand Lights [36], Colight [37], Metalight [39], Attendlight [40], and others. We have also planned to extend this work to other CPS platforms, such as the control of UAV swarm and large scale power grid. (For UAV swarm simulations, we will consider Microsoft Airsim platform, and for power grid simulations, we will consider Grid2Op from RTE France.)

---

> > ### Author Response · Authors · 2021-08-24
> > **Sincerely expecting further discussions from reviewer SpfT**
> >
> >
> > Dear Reviewer SpfT,
> >
> > We thank reviewer SpfT’s time for the review, and we really hope to have a further discussion with reviewer SpfT to see if our response solves the concerns.
> >
> > We have supplemented explanations about your concerns in the rebuttal. For example, as we pointed out that our research object (CPS) is fundamentally different from NLP in that NLP models do prevent “cheating” but don’t have spatial temporal prior, while in CPS we don’t prevent “cheating” (since there’s no future data, hence there’s no need to do so) while we do apply spatial temporal prior.
> >
> > Meanwhile, as reviewer SpfT suggested, we have riched the literature review on masks; and in response to the doubt of the individual component contribution, we have added new ablation study experiment series to demonstrate them.
> >
> > We genuinely hope reviewer SpfT could kindly check our response and make this work known by more researchers together. Thanks a lot!
> >
> > Best wishes,
> >
> > Authors

---

> > > ### Comment · Reviewer_SpfT · 2021-08-31
> > > **follow up**
> > >
> > > Thanks for your detailed response. I agree that the motivation and implementation for masking in this paper is certainly different from those considered in the NLP literature. My main concern was whether the contribution here was substantial enough or whether it should be considered a tweak on existing masking schemes. I think the response does a decent job arguing for the former.
> > >
> > > I am still concerned about the significance of the approach. In the "significance" portion of my original review, I brought up concerns about the bespoke dataset/simulation platform and lack of comparison on well known benchmark tasks. I appears that Reviewer qqzv shares these concerns that the results may only be interesting to a small audience. Your response mentions plans to perform UAV swarm and power grid simulations - do you have any updates on that? Are there existing techniques that serve as good benchmarks for those platforms?
> > >
> > > Overall, given the updates to the exposition but lack of clarity on broader significance of the work, I have raised my score to a 5.

---

> > > > ### Author Response · Authors · 2021-08-31
> > > > **Follow up**
> > > >
> > > > Thank you for your response and acknowledgment of our previous response.
> > > >
> > > > We have verified the applicability of the proposed architecture to the two new CPS tasks (robot (UAV) swarm control and power grid control) with two groups of experts in these domains. According to the experts in robot swarm control, the key requirement to collectively accomplish the control target is to address the tension between local perceptions and the global objective, and DePT's combination of global view and physical locality prior provides a practicable solution. According to the experts in power grid control, the difficulty of managing large scale power grid mainly dwells on the massive scale of the state and action space, and the delayed propagation of power supply/demand back and forth between generators, substations, and the loads; DePT's free of large state/action space and the built-in delayed propagation prior offers promising perspectives to alleviate these difficulties.
> > > >
> > > >
> > > > Based on the feedback from the two groups of experts in the robot swarm and power grid control domains, we believe DePT can be fairly compared/validated with models in these domains, and can bring broad impact to the community. We are also in the course of implementing DePT in these two systems.
> > > >
> > > >
> > > > The common simulation platform for robot swarm is Microsoft Airsim, and common benchmarks include:
> > > >
> > > >
> > > > [a1] Tolstaya, Ekaterina, et al. "Learning decentralized controllers for robot swarms with graph neural networks." Conference on robot learning. PMLR, 2020.
> > > >
> > > > [a2] Teruel, Enrique, Rosario Aragues, and Gonzalo López-Nicolás. "A distributed robot swarm control for dynamic region coverage." Robotics and Autonomous Systems 119 (2019): 51-63.
> > > >
> > > > [a3] Hu, Ting-Kuei, et al. "VGAI: End-to-end learning of vision-based decentralized controllers for robot swarms." ICASSP 2021-2021 IEEE International Conference on Acoustics, Speech and Signal Processing (ICASSP). IEEE, 2021.
> > > >
> > > > [a4] Gama, Fernando, et al. "Convolutional neural network architectures for signals supported on graphs." IEEE Transactions on Signal Processing 67.4 (2018): 1034-1049.
> > > >
> > > >
> > > >
> > > > The common platform for power grid control is high fidelity simulator Grid2Op (https://github.com/rte-france/Grid2Op), and benchmarks include:
> > > >
> > > > [b1] Yoon, Deunsol, et al. "Winning the L2RPN Challenge: Power Grid Management via Semi-Markov Afterstate Actor-Critic." International Conference on Learning Representations. 2020.
> > > >
> > > > [b2] Duan, Jiajun, et al. "Deep-reinforcement-learning-based autonomous voltage control for power grid operations." IEEE Transactions on Power Systems 35.1 (2019): 814-817.
> > > >
> > > > [b3] Marot, Antoine, et al. "Learning to run a Power Network Challenge: a Retrospective Analysis." arXiv preprint arXiv:2103.03104 (2021).
> > > >
> > > > [b4] https://github.com/rte-france/l2rpn-baselines
> > > >
> > > >
> > > > In comparison to DePT, [a1,a2,a3,a4,b1] used Graph Neural Network models to achieve decentralized processing and message passing/aggregation, while [b2] used centralized method (MLP).
> > > >
> > > > We wish our response could be helpful in addressing your concerns.
> > > >
> > > >
> > > > Sincerely,
> > > >
> > > > Authors

---

### Decision · Program_Chairs · 2021-09-27

**Decision:**

Accept (Poster)

**Comment:**

This work has ben extensively discussed by reviewers, and the AC (myself) has also stepped in to review the paper.

In general this work studies an application of model-free RL, together with a newly proposed transformer architecture with a cone-shaped attention to handle information flow in the temporal-spatial sense. This paper is more of an applied work, with not much algorithmic/theoretical contribution. Still I believe the problem studied is refreshing to the RL community, and the results which are decently good (though not super impressive) when compared with SOTA, demonstrates initial success of applying this ML technique to a more realistic problem. Most importantly, this also shows how transformers can be applied to problems beyond language and small-scale control problems in MuJoCo with novel features in the architectural design, which is new on both transformer design research and the CPS application. The paper is generally well-written, with the newly proposed transformer architecture clearly stated and discussed. On the overall, it's easy to follow and intuitive.

Several reviewers express concerns on the significance of this work, issues in evaluation, and the value to the broader community. I agree these points are valid, but given the new ideas proposed for transformer-based architecture and the applications to multi-agent systems with these techniques are quite interesting, I think the benefits outweighs the drawback. On the overall, I'd recommend acceptance for this paper.